# Urinary Titin N-Fragment Evaluation in a Randomized Controlled Trial of Beta-Hydroxy-Beta-Methylbutyrate for Acute Mild Trauma in Older Adults

**DOI:** 10.3390/nu13030899

**Published:** 2021-03-10

**Authors:** Hidehiko Nakano, Hideki Hashimoto, Masaki Mochizuki, Hiromu Naraba, Yuji Takahashi, Tomohiro Sonoo, Yujiro Matsuishi, Yasuhiro Ogawa, Nobutake Shimojo, Yoshiaki Inoue, Kensuke Nakamura

**Affiliations:** 1Department of Emergency and Critical Care Medicine, Hitachi General Hospital, Hitachi, Ibaraki 317-0077, Japan; hidehashimoto-tky@umin.ac.jp (H.H.); kurakan72@gmail.com (M.M.); nrbhrm@gmail.com (H.N.); yuji.mail@icloud.com (Y.T.); sonopy77@gmail.com (T.S.); mamashockpapashock@yahoo.co.jp (K.N.); 2Department of Emergency and Critical Care Medicine, Faculty of Medicine, University of Tsukuba, Tsukuba, Ibaraki 305-8575, Japan; 0326yujiro@gmail.com (Y.M.); ogawa_yasuhiro@hotmail.com (Y.O.); nokeshimojo@yahoo.co.jp (N.S.); yinoue@md.tsukuba.ac.jp (Y.I.)

**Keywords:** titin, beta-hydroxy-beta-methylbutyrate, muscle injury, sarcopenia, frailty

## Abstract

The effects of beta-hydroxy-beta-methylbutyrate (HMB) complex administration and the significance of titin, a biomarker of muscle injury, in elderly minor trauma patients in acute phase has not been established. In this single-center, randomized controlled study, trauma patients aged ≥ 70 years with an injury severity score < 16 were included. Titin values on days 1 and 3 were measured and the intervention group received HMB complex (2.4 g of HMB + 14 g of glutamine + 14 g of arginine) and the control group received glutamine complex (7.2 g of protein including 6 g of glutamine). The cross-sectional area of the rectus femoris (RFCSA) on ultrasound, grip strength, and the Barthel Index were assessed on the first day of rehabilitation and after 2 weeks. We analyzed 24 HMB and 25 control participants. Titin values on day 3 correlated with grip strength (*r* = −0.34, *p* = 0.03) and the Barthel Index (*r* = −0.39, *p* = 0.01) at follow-up. HMB complex supplementation had no effect on the RFCSA (2.41 vs. 2.45 cm^2^, *p* = 0.887), grip strength (13.3 vs. 13.1 kg, *p* = 0.946), or the Barthel Index (20.0 vs. 50.0, *p* = 0.404) at follow-up. Titin values might associate with subsequent physical function. Short-term HMB complex supplementation from acute phase did not ameliorate muscle injury.

## 1. Introduction

Elderly trauma patients often have difficulty reintegrating into society due to poor physical function after acute care, even if the injury is mild [1]. This vulnerability to stress in the elderly is recognized as frailty [2]. The concept of post-acute-care syndrome (PACS) has been proposed to describe the decline in physical function after acute care in the elderly [3], and physical function is considered to be an important outcome of acute care.

Since there are no biomarkers of physical function decline after acute care, difficulties have been associated with real-time assessments and interventions. Titin, a component protein of the sarcomeres of striated muscles, has recently been attracting increasing attention as a new marker of muscle disorders, and measurements of the urinary excretion of the N-terminal fragment of titin (N-titin) have been established [4]. We previously reported that urinary N-titin was markedly elevated in patients admitted to an intensive care unit (ICU) as a result of muscle injury under critical conditions and was associated with post-treatment muscle strength and physical function decline [5]. However, it currently remains unclear whether urinary N-titin is a predictive marker of muscle strength or physical function decline, even in mildly ill patients.

Muscle mass loss, known as sarcopenia, is closely related to physical frailty, and, thus, a strong relationship exists between frailty and the nutritional status [6,7]. Although anabolic resistance has been implicated in muscle mass loss in the elderly [8], this may be prevented in healthy elderly individuals by the administration of sufficient amounts of protein, including leucine [9,10]. Beta-hydroxy-beta-methylbutyrate (HMB), a leucine metabolite, activates the mechanistic target of rapamycin pathway to promote muscle protein synthesis and inhibits the ubiquitin-proteasome pathway to reduce muscle protein breakdown [11], and has been shown to increase muscle mass in the elderly [12]. A previous study demonstrated that the administration of HMB alone may improve muscle strength and quality in elderly individuals who did not perform resistance training [13]. However, it currently remains unclear whether the administration of HMB from the acute phase of disease or trauma prevents PACS.

Therefore, we measured urinary N-titin in the acute phase and investigated its usefulness as a predictive marker of muscle injury and physical function decline in elderly patients aged 70 years or older hospitalized for minor trauma. We also conducted an open-label randomized controlled trial to investigate the effects of HMB on muscle mass, strength, and physical function.

## 2. Materials and Methods

### 2.1. Patient Selection

After Ethics Committee approval (2020-37), this single-center, open-label, randomized controlled trial was conducted at Hitachi General hospital. The present study is registered as UMIN000040292. Patients aged 70 years or older who were admitted to our department for minor trauma between November 2019 and June 2020 and who were expected to be hospitalized for at least 3 days were included. Minor trauma was defined by an injury severity score (ISS) <16. Patients with renal dysfunction, on dialysis or with a serum creatinine level of 3 or higher, unable to perform enteral feeding, or who did not consent to the study were excluded. Consent was obtained from the patient or a surrogate before the initiation of the present study.

### 2.2. Randomization and Allocation

Randomization was performed using the block randomization method. Blocks of 2, 4, 6, and 8 individuals were randomly generated using random numbers, and random numbers were then generated within each block and divided into two groups by odd and even numbers when sorted by numbers.

### 2.3. Sample Size Estimation

A power analysis was conducted using software (G*Power 3 for Mac; Heinrich Heine University, Dusseldorf, Germany) during the planning phase of the present study. The effect size was estimated from our previous study [14], in which the reduction observed in femoral muscle volume in patients with sequential organ failure assessment (SOFA) score <10 was 14.0 ± 6.9% in the control group and 8.7 ± 6.4% in the HMB administration group. We calculated the sample size at the 5% level of significance and 80% power, which indicated the need for 21 participants in each group. Therefore, 25 subjects were recruited in each group to account for potential dropouts.

### 2.4. Protocol

The first morning after hospitalization was set as day 1, and randomized assignment was performed on that morning. Spot urine samples were collected for the measurement of urinary N-titin on days 1 and 3. Urine samples were stored at −20 °C for later analysis. Frozen urine samples were transported to University of Tsukuba and measured using an enzyme-linked immunosorbent assay with an assay kit (#27900 Human Titin N-Fragment Assay Kit; Immuno-Biological Laboratories, Fujioka, Japan). All samples were assayed in duplicate and the average value was used.

Both groups were given early rehabilitation and active weaning. The HMB group received Abound^®^ (Abbott Japan LLC, Tokyo, Japan) containing 1.2 g of HMB + 7 g of glutamine + 7 g of arginine per package and the control group received GFO^®^ (Otsuka Pharmaceutical Factory, Inc., Tokushima, Japan) containing 3.6 g of protein (including 3 g of glutamine) per package. Both groups were administered 2 packages per day for 14 days. Nasogastric tubes for feeding were inserted for patients with swallowing difficulties.

The initial assessment was performed at the time of the first rehabilitation after admission (days 0–2). We evaluated the cross-sectional area of the rectus femoris (RFCSA) as a muscle mass evaluation, grip strength as muscle strength, and the Barthel Index as physical function. Ultrasound images were used to evaluate RFCSA, with SonoSite Edge and the 6-MHz linear probe HFL38xi (FUJIFILM Medical Co., Ltd., Tokyo, Japan). Ultrasound evaluations were performed by trained physical therapists. RFCSA was measured at the midpoint of the superior anterior iliac spine and superior edge of the patella. A linear probe was applied in the axial direction using a sufficient amount of gel to avoid deforming RF. The distance from the upper edge of the patella to the measurement point was recorded so that it could be measured at the same location in the follow-up. Grip strength was measured and recorded once on each side using a digital grip strength meter TKK-5401 (Takei Kiki Kogyo Co., Ltd., Tokyo, Japan), and the larger value was used for analysis. The Barthel Index prior to injury was assessed by interviewing the patients or their family. We evaluated SOFA scores and acute physiology and chronic health evaluation (APACHE) II scores on day 1, and also measured SOFA every day. ICU admission and adjunctive treatments, such as mechanical ventilation and renal replacement therapy, were also recorded. If oral intake was not possible, enteral nutrition was administered, and nutrition was calculated from the amount administered. If the patient was taking food orally, nutrient intake data were assessed by the hospital nutritionist using the hospital ready reckoners and the recorded food intake. Intravenous fluids were also included in the nutritional calculations.

Follow-up assessments were performed between days 14 and 21. Inpatients were assessed during hospitalization, while patients who were already discharged were assessed on an outpatient basis.

### 2.5. Outcome Evaluation

The primary endpoint was cross-sectional area of muscle, assessed by RFCSA, grip strength and the Barthel Index in the follow-up. Secondary endpoints were urinary N-titin levels on days 1 and 3. In the N-titin evaluation, we used N-titin/Cre (pmol/mgCre), which was calculated by dividing the spot urine N-titin concentration (pmol/L) by the spot urine creatinine concentration (mg/dL), multiplied by 10, as described in a previous study [5].

We also performed a subgroup analysis that excluded critically ill patients. The mean of the maximum SOFA of surviving patients in the ICU in a previous study was 6.7 [15], and, thus, maximum SOFA of 7 was set as the cut-off.

To investigate the usefulness of urinary N-titin as a marker of muscle injury, we compared patients with maximum SOFA <7 with those with maximum SOFA ≥7 to confirm the relationship between severity and muscle injury. We also examined the relationships between N-titin/Cre and RFCSA, grip strength, and the Barthel Index in the follow-up.

### 2.6. Statistical Analysis

We evaluated the normality of the distribution by the Shapiro–Wilk test. Parametric continuous variables were expressed as the mean ± standard deviation, and non-parametric continuous variables as medians (interquartile range). Comparisons between the two groups were performed using the Student’s *t*-test for parametric continuous variables, the Mann–Whitney U test for non-parametric variables, and the chi-squared or Fisher’s exact test for categorical variables. The relationship between two variables was calculated by Spearman’s correlation coefficient. A *p*-value < 0.05 was considered to indicate a significant difference. Statistical analyses were performed using R (version 3.6.1., R Foundation for Statistical Computing, Vienna, Austria).

## 3. Results

Among 1340 patients admitted during the study period, 336 were admitted for trauma, including 98 patients aged 70 years or older. Following the exclusion of 48 patients who met the exclusion criteria, 50 patients were enrolled (Figure 1). Excluded patients included 19 with ISS ≥ 16, 17 discharged early, 6 with renal dysfunction, and 6 unable to provide consent.

Patient characteristics are shown in Table 1. Patients had a mean age of 84.4 years, and the degree of trauma was a median ISS of 9.0. In terms of severity, median SOFA on day 1 was 2.0, median maximum SOFA was 3.0, and median APACHE II was 10.0. One patient in both groups died during hospitalization, and three patients in the HMB group were admitted to the ICU and required mechanical ventilation. The length of the hospital stay was slightly longer in the HMB group (19.5 vs. 15.0 days, *p* = 0.055). No significant differences were observed in nutritional therapy during hospitalization between the HMB and control groups, with calorie delivery of 20.9 vs. 19.5 kcal/kg (*p* = 0.626), and protein delivery of 0.9 vs. 0.8 g/kg (*p* = 0.475), respectively.

Table 2 shows the results of measurements. The follow-up assessment date was 14.7 days after the initial assessment in the HMB group and 15.1 days in the control group. RFCSA, a measure of muscle mass, did not significantly differ between the two groups in the initial assessment (2.49 vs. 2.28 cm^2^, *p* = 0.300) or in the follow-up assessment (2.41 vs. 2.45 cm^2^, *p* = 0.887), and no significant differences were observed in changes (−0.04 mm^2^ vs. 0.33 mm^2^, *p* = 0.250). Similarly, changes in grip strength (0.8 kg vs. 1.0 kg, *p* = 0.508) and the Barthel Index (−42.5 vs. −35.0, *p* = 0.311) did not significantly differ between the two groups. The value of N-titin/Cre, a biomarker of muscle injury, on day 1 (22.3 vs. 20.6 pmol/mgCre, *p* = 0.801) and changes (5.9 vs. 0.3 pmol/mgCre, *p* = 0.120) were also not significantly different, whereas that on day 3 (27.3 vs. 18.2 pmol/mgCre, *p* = 0.046) was significantly higher in the HMB group. Patients who underwent follow-up evaluation during hospitalization and those who underwent outpatient evaluation were also compared between the HMB group and the control group, respectively, but no differences were found in either group (Appendix A).

To exclude patients who became critically ill during the course of their hospitalization due to complications, a subgroup analysis of patients with maximum SOFA < 7 was performed, and the results obtained are shown in Table 3. After the exclusion of patients who became severely ill during the course of their hospitalization, no significant differences were observed in RFCSA, grip strength, the Barthel Index, or N-titin/Cre.

To confirm the impact of disease severity on muscle injury, the results of a comparison between non-severely and severely ill patients are shown in Table 4. No significant difference was observed in RFCSA. Grip strength in the follow-up was slightly weaker in the severely ill group (14.1 kg vs. 7.7 kg, *p* = 0.079). The Barthel Index in the follow-up was lower in the severely ill group (52.5 vs. 5.0, *p* = 0.048). Furthermore, N-titin/Cre on day 3 was higher in the severely ill group (19.0 pmol/mgCre vs. 38.1 pmol/mgCre, *p* = 0.026).

To compare the relationships between urinary titin and muscle mass, strength, and physical function, we examined those between N-titin/Cre and RFCSA, grip strength, and the Barthel Index in the follow-up (Figure 2). Regarding N-titin/Cre and RFCSA, no correlation was observed between N-titin/Cre on day 1 (*r* = −0.28, *p* = 0.06) or day 3 (*r* = −0.26, *p* = 0.10). Furthermore, grip strength did not correlate with N-titin/Cre on day 1 (*r* = −0.15, *p* = 0.34), whereas an inverse correlation was observed on day 3 (*r* = −0.34, *p* = 0.03). Similarly, the Barthel Index did not correlate with N-titin/Cre on day 1 (*r* = −0.22, *p* = 0.14), whereas an inverse correlation was noted on day 3 (*r* = −0.39, *p* = 0.01). Of the seven patients with grip strength and Barthel Index of 0 at the time of follow-up evaluation, six had elevated N-titin/Cre. These patients included two patients with head trauma, three with complications of infection, and one centenarian, all of whom had progressive disuse during hospitalization.

In scatter plots, the regression line is shown as a solid line, with 95% confidence intervals as shaded areas. No correlation was observed between RFCSA and N-titin/Cre on day 1 (*r* = −0.28, *p* = 0.06) or day 3 (*r* = −0.26, *p* = 0.10). Furthermore, no correlation was noted between grip strength and N-titin/Cre on day 1 (*r* = −0.15, *p* = 0.34), whereas an inverse correlation was observed on day 3 (*r* = −0.34, *p* = 0.03). Similarly, the Barthel Index did not correlate with N-titin/Cre on day 1 (*r* = −0.22, *p* = 0.14), whereas an inverse correlation was observed on day 3 (*r* = −0.39, *p* = 0.01). RFCSA = cross-sectional area of the rectus femoris; N-titin/Cre = spot urine titin N-fragment divided by spot urine creatinine multiplied by 10.

Details of the correlation coefficients of other N-titin/Cre and muscle injury endpoints are shown in Table 5. There was no significant correlation between N-titin/Cre on day 1 and muscle injury endpoints on day 1. The inverse correlations between N-titin/Cre on day3 and grip strength and Barthel Index at follow-up were stronger when only patients evaluated on admission were included.

Comparison of N-titin/Cre and urinary creatinine on day 3 showed an inverse correlation (*r* = −0.63, *p* < 0.01), but the correlation with physical function was better for N-titin/Cre (Appendix A).

## 4. Discussion

Urinary N-titin, a marker of muscle injury, inversely correlated with grip strength and the Barthel Index, but not with RFCSA. Short-term administration of 2.4 g of HMB + 14 g of glutamine + 14 g of arginine per day from acute phase did not prevent muscle mass loss, or physical function decline in elderly patients with minor trauma.

Urinary N-titin remained significantly higher on day 3 in patients who became critically ill and correlated with grip strength and the Barthel Index in the follow-up. Normal N-titin/Cre values were previously reported to be 1.09–7.09 (pmol/mgCre) in healthy adults [16], and 50.6 [24.0–111.7] pmol/mgCre in ICU-admitted patients in our previous study [5]. N-titin/Cre values in the present study were 20.7 [15.2–31.2] pmol/mgCre. A previous study that investigated changes in N-titin/Cre after exercise in healthy individuals reported that values reached a maximum (27 pmol/mgCre) after approximately 10 h and returned to normal (5 pmol/mgCre) after approximately 30 h [16]. It was reported that N-titin decreases during the initial period of bed rest but increases after 2–3 weeks in healthy volunteers [17]. Therefore, N-titin/Cre values on day 1 were considered to be affected by trauma, while high levels of N-titin/Cre on day 3 may indicate the emergence of persistent muscle injury due to stress-related inflammation. We previously reported that N-titin/Cre ≥100 pmol/mgCre in critically ill patients indicated the development of ICU acquired weakness [5]. We also showed that urinary N-titin was associated with muscle strength, but not muscle volume, which was consistent with the present results. In addition to critically ill patients, mildly ill patients may benefit from an evaluation of muscle injury using measurements of urinary N-titin levels. By assessing N-titin, it may be possible to identify at-risk patients and intervene from earlier with exercise and nutritional therapy to prevent physical function decline. N-titin assessment may also allow us to evaluate the effect of the optimal exercise and nutritional therapy, or specific treatment such as drugs with anti-inflammatory effects or protease inhibitors. Furthermore, muscle injury or muscle weakness did not necessarily correlate with muscle mass in the acute post-traumatic period; therefore, muscle injury and muscle mass may need to be separately assessed.

Studies on the long-term administration of HMB to healthy elderly subjects reported muscle mass gain after 12–48 weeks of treatment [18,19]. Stout et al. evaluated a combination of HMB and resistance training for 24 weeks in healthy elderly participants and found that, with exercise, muscle mass increased with or without HMB, whereas without exercise, muscle mass only increased in the HMB group [13]. The effects of a short-term nutritional intervention on muscle mass were previously reported to increase lean body mass in malnourished older adults with an additional 30 g of protein for 10 days [20]. Deutz et al. performed a 10-day bed rest study on 19 healthy older adults aged 60–76 years, and showed that a dose of 3 g/day of HMB prevented muscle mass loss [21]. Although previous studies demonstrated the benefits of short-term administration, HMB supplementation did not exert any beneficial effects during this time period.

In the present study, ICU admission only occurred in the HMB group, suggesting that severe complications during hospitalization affected outcomes. The excessive administration of amino acids in the acute phase to critically ill patients generally suppresses autophagy, and these doses need to be reduced in the hyperacute phase [22]. We previously demonstrated the preventive effects of the combination of HMB and electrical stimulation therapy on muscle mass loss in patients with SOFA ≤ 10 only [14], which also suggests that the administration of HMB is not significantly beneficial until after the acute phase. Moreover, the HMB group simultaneously received 14 g of arginine per day, which may exert more negative effects than HMB because previous studies indicated that arginine exerted adverse effects in septic patients [23,24]. However, a subgroup analysis of patients who did not become severely ill did not show any significant differences, and the present study did not show any benefits of HMB.

In addition, protein delivery during admission, including supplements, in both groups was only approximately 0.9 g/kg, which was insufficient. A protein intake of 1.0–1.3 g/kg is recommended due to anabolic resistance in the elderly [25]. In addition, it is known that muscle protein breakdown is accelerated by disuse and inflammation [26], and protein requirements increase after injury. Protein requirements in the acute phase of critical illness are recommended to be 1.3 g/kg or higher, and protein intake should be further increased to 2.0–2.5 g/kg during recovery phase [27]. In a study on short-term HMB administration conducted by Deutz et al., older adults were given an adequate protein delivery of 1.0 g/kg or more [21]. The lack of an adequate intake of essential amino acids (particularly branched chain amino acids), which are necessary for muscle protein synthesis, may also have influenced the lack of effects of HMB intake.

The present study has some limitations. First, although not significantly different, only the HMB group had patients admitted to the ICU, and there was a tendency for more patients to be evaluated for follow-up during hospitalization. Even in a randomized controlled trial, selection bias may have occurred and may have affected the results. Second, there was a lack of well-protocolized nutritional therapy and rehabilitation. As a result, adequate protein administration was not achieved. Furthermore, exercise may have been inadequate because we did not perform holiday rehabilitation or neuromuscular electrical stimulation in cases in which weaning was not possible. Additionally, we were unable to record the amount of exercise and nutrient intake after discharge from the hospital. Therefore, HMB intake may have been inadequate after discharge from the hospital. Another limitation was the measurement bias of the ultrasound. Although the examiner was sufficiently trained for the ultrasound assessment, measurement bias may have affected the results obtained. Lastly, we only evaluate N-titin in acute phase. In future study, we need to evaluate N-titin levels in post-acute phase.

## 5. Conclusions

In patients older than 70 years of age hospitalized for minor trauma, 2 weeks of HMB 2.4 g + 14 g of glutamine + 14 g of arginine did not ameliorate muscle mass loss, muscle weakness, or physical function decline. Urinary N-titin correlated with muscle strength and physical function, indicating its potential as a marker of muscle injury, even in mildly ill patients.

## Figures and Tables

**Figure 1 nutrients-13-00899-f001:**
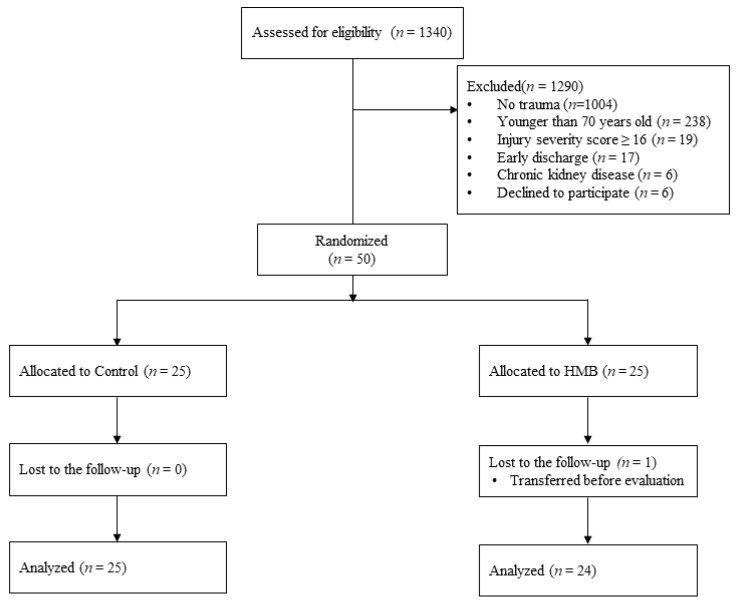
Patient selection. HMB = beta-hydroxy-beta-methylbutyrate.

**Figure 2 nutrients-13-00899-f002:**
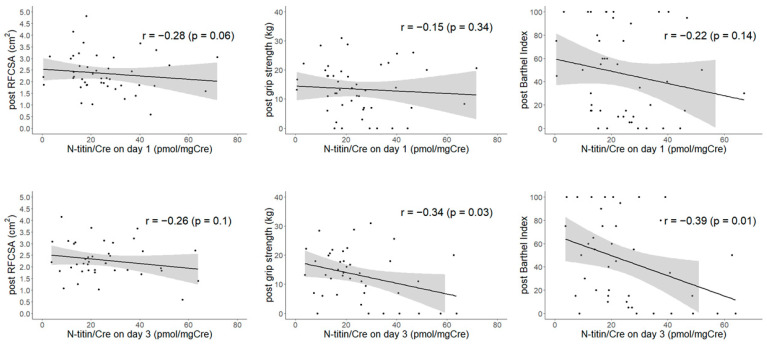
Scatter plots between N-titin/Cre and muscle injury markers.

**Table 1 nutrients-13-00899-t001:** Basic characteristics and comparisons between groups.

	HMB	Control	*p*
*n*	24	25	
Age, years	83.8 ± 5.2	84.9 ± 7.0	0.523
Male, *n* (%)	12 (50.0)	10 (40.0)	0.677
Body height, cm	152.2 ± 10.3	152.6 ± 8.3	0.866
Body weight, kg	50.4 ± 11.4	51.6 ± 8.8	0.708
SOFA initial	1.0 (0.8 to 3.0)	2.0 (1.0 to 3.0)	0.438
SOFA maximum	3.0 (2.0 to 5.2)	4.0 (3.0 to 4.0)	0.598
APACHE II	10.0 (8.0 to 11.2)	10.0 (8.0 to 16.0)	0.644
ISS	4.0 (4.0 to 9.0)	9.0 (4.0 to 9.0)	0.152
CCI	2.0 (1.0 to 3.2)	1.0 (1.0 to 2.0)	0.431
Length of hospitalization, day	19.5 (13.8 to 39.0)	15.0 (5.0 to 25.0)	0.055
ICU admission, *n* (%)	3 (12.5)	0 (0.0)	0.219
Duration of ICU admission	0.7 ± 2.2	0.0 ± 0.0	0.143
Death, *n* (%)	1 (4.2)	1 (4.0)	1
MV use, *n* (%)	3 (12.5)	0 (0.0)	0.219
Duration of MV use, day	0.4 ± 1.4	0.0 ± 0.0	0.199
RRT use, *n* (%)	1 (4.2)	0 (0.0)	0.984
Calorie delivery, kcal *	20.9 ± 10.3	19.5 ± 7.9	0.626
Protein delivery, g *	0.9 ± 0.4	0.8 ± 0.4	0.475

Data were shown as means ± SD or medians (IQR). *: value during hospitalization. HMB = beta-hydroxy-beta-methylbutyrate; SOFA = sequential organ failure assessment; APACHE = acute physiology and chronic health evaluation; ISS = Injury severity score; CCI = Charlson comorbidity index, ICU = intensive care unit; MV = mechanical ventilation; RRT = renal replacement therapy.

**Table 2 nutrients-13-00899-t002:** Results and comparisons between groups.

	HMB	Control	*p*
*n*	24	25	
Follow-up evaluation day	14.7 ± 3.1	15.1 ± 3.0	0.605
RFCSA			
pre, cm^2^	2.49 ± 0.77	2.28 ± 0.59	0.3
post, cm^2^	2.41 ± 0.95	2.45 ± 1.09	0.887
change, cm^2^	−0.04 (−0.58 to 0.50)	0.33 (−0.10 to 0.53)	0.25
Grip strength			
pre, kg	12.5 ± 7.3	10.4 ± 7.0	0.314
post, kg	13.3 ± 9.1	13.1 ± 8.9	0.946
change, kg	0.8 (−1.0 to 3.4)	1.0 (0.0 to 3.0)	0.508
Barthel Index			
pre	100.0 (93.8 to 100.0)	100.0 (95.0 to 100.0)	0.71
post	20.0 (10.0 to 75.0)	50.0 (20.0 to 90.0)	0.404
change	−42.5 (−81.2 to −23.8)	−35.0 (−60.0 to −10.0)	0.311
N-titin/Cre			
Day 1, pmol/mgCre	22.3 (15.9 to 29.7)	20.6 (12.7 to 34.2)	0.801
Day 3, pmol/mgCre	27.3 (19.1 to 39.1)	18.2 (12.1 to 25.4)	0.046 *
change, pmol/mgCre	5.9 (−0.4 to 16.5)	0.3 (−4.3 to 10.2)	0.12

Data were shown as means ± SD or medians (IQR). *: *p* < 0.05. HMB = beta-hydroxy-beta-methylbutyrate; RFSCA = cross-sectional area of the rectus femoris, N-titin/Cre = spot urine titin N-fragment divided by spot urine creatinine multiplied by 10.

**Table 3 nutrients-13-00899-t003:** Results of the subgroup analysis in patients with maximum SOFA < 7.

	HMB	Control	*p*
*n*	19	23	
Age, years	84.1 ± 4.8	84.7 ± 7.1	0.776
Male, *n* (%)	8 (42.1)	10 (43.5)	1
SOFA maximum	2.0 (2.0 to 3.0)	3.0 (2.5 to 4.0)	0.091
APACHE II	9.0 (8.0 to 11.0)	10.0 (8.0 to 12.5)	0.359
ISS	4.0 (4.0 to 9.0)	9.0 (5.0 to 9.0)	0.089
CCI	2.0 (1.0 to 3.5)	1.0 (1.0 to 2.5)	0.315
Length of hospitalization, day	19.0 (13.5 to 32.0)	15.0 (4.5 to 25.0)	0.069
ICU admission, *n* (%)	0.0 (0.0)	0.0 (0.0)	NA
Death, *n* (%)	0.0 (0.0)	0.0 (0.0)	NA
Follow-up evaluation day	14.1 ± 2.5	15.1 ± 2.9	0.257
RFCSA			
pre, cm^2^	2.38 ± 0.75	2.33 ± 0.59	0.807
post, cm^2^	2.32 ± 0.88	2.56 ± 1.06	0.427
change, cm^2^	0.05 (−0.52 to 0.46)	0.33 (−0.09 to 0.57)	0.216
Grip strength			
pre, kg	13.2 ± 7.2	10.5 ± 6.8	0.222
post, kg	14.8 ± 8.8	13.5 ± 8.8	0.641
change, kg	1.0 (−0.7 to 3.5)	1.4 (0.0 to 3.5)	0.63
Barthel Index			
pre	100.0 (90.0 to 100.0)	100.0 (95.0 to 100.0)	0.701
post	30.0 (15.0 to 75.0)	55.0 (25.0 to 92.5)	0.543
change	−35.0 (−77.5 to −22.5)	−35.0 (−60.0 to −7.5)	0.751
N-titin/Cre			
Day 1, pmol/mgCre	23.3 (15.8 to 37.5)	20.6 (12.8 to 30.0)	0.693
Day 3, pmol/mgCre	25.4 (18.8 to 37.5)	17.0 (11.5 to 23.9)	0.077
change, pmol/mgCre	5.7 (−1.1 to 11.5)	−0.5 (−5.0 to 4.6)	0.135

Data were shown as means ± SD or medians (IQR). HMB = beta-hydroxy-beta-methylbutyrate; NA = not applicable; SOFA = sequential organ failure assessment; APACHE = acute physiology and chronic health evaluation; ISS = Injury severity score; CCI = Charlson comorbidity index, ICU = intensive care unit; RFSCA = cross-sectional area of the rectus femoris; N-titin/Cre = spot urine titin N-fragment divided by spot urine creatinine multiplied by 10.0.

**Table 4 nutrients-13-00899-t004:** Comparisons between patients with maximum SOFA < 7 and those with ≥7.

	MAX SOFA < 7	MAX SOFA ≥ 7	*p*
*n*	42	7	
Age, years	84.4 ± 6.1	83.9 ± 6.6	0.829
Male, *n* (%)	18 (42.9)	4 (57.1)	0.769
SOFA maximum	3.0 (2.0 to 4.0)	8.0 (7.5 to 10.5)	<0.001 *
APACHE II	10.0 (8.0 to 11.8)	18.0 (14.0 to 23.0)	0.001 *
ISS	8.5 (4.0 to 9.0)	9.0 (4.0 to 9.0)	0.917
CCI	2.0 (1.0 to 3.0)	1.0 (0.5 to 1.5)	0.249
Length of hospitalization, day	15.5 (7.2 to 27.0)	25.0 (20.0 to 60.5)	0.109
ICU admission, *n* (%)	0 (0.0)	3 (42.9)	<0.001 *
Death, *n* (%)	0 (0.0)	2 (28.6)	0.012 *
Follow-up evaluation day	14.6 ± 2.8	16.4 ± 4.2	0.15
RFCSA			
pre, cm^2^	2.35 ± 0.66	2.58 ± 0.85	0.411
post, cm^2^	2.45 ± 0.98	2.33 ± 1.29	0.778
change, cm^2^	0.15 (−0.35 to 0.55)	−0.37 (−0.84 to 0.43)	0.338
Grip strength			
pre, kg	11.7 ± 7.1	9.3 ± 8.2	0.41
post, kg	14.1 ± 8.7	7.7 ± 8.3	0.079
change, kg	1.2 (−0.3 to 3.7)	0.0 (−4.0 to 0.5)	0.133
Barthel Index			
pre	100.0 (91.2 to 100.0)	100.0 (100.0 to 100.0)	0.391
post	52.5 (15.0 to 78.8)	5.0 (0.0 to 30.0)	0.048 *
change	−35.0 (−67.5 to −12.5)	−85.0 (−97.5 to −35.0)	0.091
N-titin/Cre			
Day 1, pmol/mgCre	22.1 (15.4 to 33.1)	18.4 (16.7 to 26.1)	0.702
Day 3, pmol/mgCre	19.1 (13.6 to 28.9)	38.1 (29.2 to 46.8)	0.03 *
change, pmol/mgCre	1.3 (−4.8 to 8.7)	18.7 (13.9 to 22.8)	0.013 *

Data were shown as means ± SD or medians (IQR). *: *p* < 0.05. HMB = beta-hydroxy-beta-methylbutyrate; NA = not applicable; SOFA = sequential organ failure assessment; APACHE = acute physiology and chronic health evaluation; ISS = Injury severity score; CCI = Charlson comorbidity index, ICU = intensive care unit; RFSCA = cross-sectional area of the rectus femoris; N-titin/Cre = spot urine titin N-fragment divided by spot urine creatinine multiplied by 10.0.

**Table 5 nutrients-13-00899-t005:** Correlation coefficients between N-titin/Cre and RFCSA, grip strength and Barthel Index.

	N-Titin/Cre on Day 1			N-Titin/Cre on Day 3			N-Titin/Cre Change		
	Overall	Outpatient	Inpatient	Overall	Outpatient	Inpatient	Overall	Outpatient	Inpatient
RFCSA									
pre, cm^2^	−0.12	−0.34	−0.04	−0.08	−0.47	0.09	0.03	−0.11	0.07
post, cm^2^	−0.28	−0.31	−0.29	−0.26	−0.38	−0.21	−0.24	−0.45	−0.19
change, cm^2^	−0.25	−0.3	−0.23	−0.23	−0.29	−0.18	−0.22	−0.4	−0.17
Grip strength									
pre, kg	−0.17	0.4	−0.3	−0.26	0.3	−0.39	−0.31	−0.21	−0.25
post, kg	−0.15	−0.04	−0.14	−0.34 *	0.29	−0.47 *	−0.34 *	0.17	−0.34
change, kg	−0.04	−0.48	0.16	−0.13	0.02	−0.05	−0.07	0.31	−0.05
Barthel Index									
pre	−0.21	−0.23	−0.22	−0.31 *	−0.08	−0.35	−0.32 *	−0.51	−0.22
post	−0.22	−0.19	−0.28	−0.39 *	−0.02	−0.47 *	−0.38 *	−0.34	−0.27
change	−0.08	−0.19	−0.03	−0.27	−0.11	−0.12	−0.22	−0.29	−0.09

*: *p* < 0.05. N-titin/Cre = spot urine titin N-fragment divided by spot urine creatinine multiplied by 10.0; RFSCA = cross-sectional area of the rectus femoris.

## Data Availability

The data presented in this study are available on request from the corresponding author. The data are not publicly available due to restrictions from the Ethics Committee.

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
