# Peer review of "Urinary Titin N-Fragment Evaluation in a Randomized Controlled Trial of Beta-Hydroxy-Beta-Methylbutyrate for Acute Mild Trauma in Older Adults"

_nutrients, 2021, doi:10.3390/nu13030899_

Round 1

Reviewer 1 Report

Thank you for the opportunity to review this manuscript titled: ‘Urinary titin N-fragment evaluation in a randomized controlled trial of beta-hydroxy-beta-methylbutyrate for acute mild trauma in older adults by Nakano et al. The primary aim of this study was to investigate the effects of HMB on muscle mass, strength, and physical function in elderly patients admitted to hospital for minor trauma. The manuscript was generally well written with the overall findings relevant to the field of gerontology and geriatrics. However, I have a few concerns which should be addressed (perhaps in the discussion) by the authors:

  1. I wonder if your results are overstated (particularly lines 222-25) – specifically how much change are you likely to see in 21-days (without an exercise intervention) in the primary outcomes reported? Secondly, it is potentially problematic that some follow-up assessments were undertaken during hospitalization (patients still admitted) whilst others were undertaken on patients already discharged. As such, patients still admitted as inpatients, and being relatively immobile, may have experienced poorer outcomes and therefore performed poorly in these assessments?

Minor comments are as follows:    

Line 82 (sample size estimation) please write out SOFA in full before it is abbreviated. It is written out in full then abbreviated in line 108 which I assume was overlooked by the authors.

The authors should report greater detail in the methodology with respect to muscle strength and nutritional analysis. For example, I am assuming grip strength was assessed using a hand-held dynamometer (equipment?). What protocol was used? For example, was maximal hand-grip strength used or the mean of multiple measures. Furthermore, please be more specific when stating nutrient intake was calculated – which nutrients e.g. energy and macronutrients? How as it calculated, e.g. food composition database?

In line 115: ‘We measured RFCSA, grip strength, and the Barthel Index again’. The authors have essentially already described this in the previous two lines by stating follow-up assessment occurred between days 14 and 21. Please consider removing.

In line 123 of the methodology, please consider revising to state that the primary endpoint was cross-sectional area of muscle, as assessed by RFCSA (not muscle mass)

In lines 211 and 213 of the results please state that these were inverse correlations (e.g. N-titin/Cre and grip and hand-grip strength on day 3 and  N-titin/Cre and Barthel Index on day 3). This should also be reported in the Figure 2 footnotes.

In lines 245-47 of the discussion, whilst I agree with the authors that protein delivery throughout hospital admission was insufficient (~0.9g/kg), in addition to the phenomenon of anabolic resistance in the elderly, the authors should also make the connection with protein requirements, critical illness and immobilization (muscle disuse).  

Reviewer 2 Report

The manuscript of Nakano et al. investigates the use of urinary titin N-fragments in elderly after mild trauma as a biomarker for physical decline and the effects of HMB supplementation as a contermeasure. Results show that values of urinary titin N-fragments significantly increase in patients with higher organ failure scores and display a trend (r=-0.34-0.39) to a negative correlation with values of post-grip strenght and post-Barthel index. Apparently no positive effect comes from HMB supplementation.

Major criticisms

  1. A major limitation of this paper comes from the mixing of two different aims, which are the predictability of urinary titin N-fragments on physical recovery from minor trauma events in the elderly and the effects of HBM treatment on the physical decline consequent to trauma and bedding/hospitalization. This appears clearly from the discussion where the concerns about HBM supplementation are presented mentioning the results obtained with urinary titin N-fragments only at the end. In my opinion, the proposal of urinary titin N-fragments as putative valuable biomarker for mild trauma in the ederly should be discussed as first, although I understand that major informations came from the re-grouping of patients' cohorts on different criteria than HBM treatment.

  2. Predictivity of urinary titin N-fragments suffers from the fact that this parameter was measured only at hospitalization day 1 and 3. Demonstration that urinary titin-N fragments remained higher until end of hospitalization would be more convincing. The rationale for such a strategy should be better clarified. The same criticism affects the regression analyses performed between values of urinary titin N-fragments and those of post-RFCSA and post-grip strenght, which were recorded after 14-21 days rehabilitation. Would the results be different when values of urinary titin N-fragments are related to those of pre-RFCSA and pre-grip strenght, which were recorded between day 0-2?

  3. Are there data in the literature that show slightly higher urinary titin N-fragments in bedridden patients?

  4. Would paired statistical analyses on day1-3 urinary titin N-fragments, pre-post RFCSA or grip strenght data, show some significant difference if assessed using the different subgroups of subjects? Regression analyses using % change (pre-post) could be also explored.

Minor changes

  1. Please, explicit abbreviations at their first use (i.e. SOFA page 2 line 82)

  2. Did duration of HBM treatment correspond to hospitalization time?

Reviewer 3 Report

The purpose of this study was to examine the efficacy of improving patient outcomes in older adult patients experiencing trauma by providing beta-hydroxy-beta-methylbutyrate supplement. N-titin urine excretion and clinical outcome measures were utilized to determine efficacy.

This reviewer commends the authors on a well-written manuscript and interesting data set. While limitations in this study (e.g., non-distinct spectrum of trauma patients) may have limited the ability to detect differences between groups, these findings provide realistic clinical insight on the utilization of HMB in treatment of trauma patients. Comments are included below.

Do the authors believe that measuring N-titin is a clinically useful marker? There appears to be a relatively weak correlation between patient outcomes and N-titin urinary excretion. Does the changes in N-titin precede worsening clinical outcomes or does it track with patient deterioration. If so, what could be done to improve patient care based on N-titin findings?

It appears that the correlation between post-grip strength/barthel index and day 3 n-titin/CRE is driven by six of the seven patients experiencing a value of 0 for post grip strength and barthel index, respectively. Can the authors clarify the cause of recording 0 values in these categories?

Values and/or units in tables are likely incorrect for RFCSA- the area of the rectus femoris was reported as 2.49mm squared. This value should likely be 100-fold higher.

What is the reasoning for normalizing N-titin to creatinine? A recent report has suggested that urinary creatinine is associated with in-hospital mortality, with lower creatinine levels having higher risk of mortality (Hessels). Is it possible that the negative association of N-titin and grip strength is linked to lower creatinine levels (therefore presenting as a higher N-titin excretion)? Can the authors provide data on creatinine levels?

Round 2

Reviewer 1 Report

Thank you for the opportunity to review the recently revised manuscript. The authors should be commended on the revisions made. I believe this manuscript is worthy of publication after two very minor revisions are made in the methodology:

  1. Please remove the word 'the' grip strength in line 112 - this should simply read 'Grip strength'
  2. It is still not clear (or appropriately described) how nutrient intake (oral intake) was analysed. Yes, nutrient intake was calculated from recorded foods intake, but how was it assessed e.g. National food composition database of nutrients, hospital ready reckoners etc. Furthermore, please reword: ‘a hospital diet with energy and macronutrients controlled by a nutritionist’. Perhaps something like the following: ‘Nutrient intake data was assessed by the hospital nutritionist using____’

    Author Response

Reviewer 2 Report

The revised version of  the manuscript improved result presentation and study limitations, addressing in depth every concern.
